# Diversity in medullary thymic epithelial cells controls the activity and availability of iNKT cells

Beth Lucas[1], Andrea J. White[1], Emilie J. Cosway[1], Sonia M. Parnell[1], Kieran D. James[1], Nick D. Jones[1], Izumi Ohigashi[2], Yousuke Takahama[3], William E. Jenkinson[1] & Graham Anderson [1✉]

The thymus supports multiple αβ T cell lineages that are functionally distinct, but mechanisms that control this multifaceted development are poorly understood. Here we examine medullary thymic epithelial cell (mTEC) heterogeneity and its influence on CD1d-restricted iNKT cells. We find three distinct mTEC[low] subsets distinguished by surface, intracellular and secreted molecules, and identify LTβR as a cell-autonomous controller of their development. Importantly, this mTEC heterogeneity enables the thymus to differentially control iNKT sublineages possessing distinct effector properties. mTEC expression of LTβR is essential for the development thymic tuft cells which regulate NKT2 via IL-25, while LTβR controls CD104[+] CCL21[+] mTEC[low] that are capable of IL-15-transpresentation for regulating NKT1 and NKT17. Finally, mTECs regulate both iNKT-mediated activation of thymic dendritic cells, and iNKT availability in extrathymic sites. In conclusion, mTEC specialization controls intrathymic iNKT cell development and function, and determines iNKT pool size in peripheral tissues.

---

[1] Institute for Immunology and Immunotherapy, University of Birmingham, Birmingham B15 2TT, UK. [2] Division of Experimental Immunology, Institute of Advanced Medical Sciences, University of Tokushima, Tokushima 770-8503, Japan. [3] Experimental Immunology Branch, National Cancer Institute, National Institutes of Health, Bethesda, MD 20892, USA. ✉email: g.anderson@bham.ac.uk

A key feature of the adaptive immune system is the development of αβT cells in the thymus. Here, specialised cortical and medullary thymic microenvironments support the step-wise maturation of self-tolerant CD4[+] and CD8[+] αβT cells that recognise peptide/major histocompatibility complex (MHC) ligands. Studies on thymic epithelial cell (TEC) heterogeneity have yielded several underlying features that determine their ability to control specific events in conventional αβT cell development[1–3]. For example, in medullary TECs (mTECs), expression of Aire together with high levels of CD80/CD86 and MHC explains the functional importance of mTEC[hi] for tolerance induction[4–6]. In addition to generating and shaping the conventional αβT cell pool, thymic microenvironments also foster other T cell lineages that are classed as 'unconventional' as they express antigen receptors that do not recognise MHC. These include CD1d-restricted invariant natural killer T cells (iNKT cells) that via their steady-state production of cytokines, including interleukin-4 (IL-4)[7–9], play important roles in the control of immune responses[10,11]. Thus, the thymus is critical for the generation of multiple αβT cell types that play key roles in the functioning of both the innate and adaptive arms of the immune system. Despite this, how the thymus controls diversity in T cell production remains poorly understood.

Here, we show that lymphotoxin β receptor (LTβR) is an essential regulator of multiple mTEC subsets within the CD80[low]MHCII[low] mTEC[low] compartment, including thymic tuft cells and the CCL21[+] subset. Importantly, we provide evidence that mTEC[low] heterogeneity is accompanied by functional specialisation. Thus, PLZF[hi]RORγt[−]Tbet[−] NKT2 are regulated by IL-25 production by thymus tuft cells, providing an explanation for the requirement for this newly described mTEC subset in iNKT cell development. Additionally, PLZF[low]RORγt[−]Tbet[+] NKT1 and PLZF[low]RORγt[+]Tbet[−] NKT17 are influenced by IL-15 transpresentation, with CCL21[+]CD104[+] mTEC[low] identified as mTECs with this function. Finally, LTβR-mediated mTEC heterogeneity enables iNKT cells to control intrathymic dendritic cell (DC) activation, and establishment of the peripheral iNKT cell pool. Collectively, our findings identify the regulation of mTEC heterogeneity by LTβR as an intrathymic mechanism that determines the availability and function of iNKT cells in both the thymus and periphery.

## Results

**Heterogeneity in the adult mTEC[low] compartment.** Traditionally, mTECs in the adult thymus are divided into MHCII[low]CD80[low] and MHCII[hi]CD80[hi] (mTEC[low]/mTEC[hi]) subsets[6,12]. While mTEC[hi] control tolerance induction through negative selection of conventional αβT cells and the generation of Foxp3[+] regulatory T cells[13,14], the functional importance of the mTEC[low] compartment remains poorly understood. Indeed, despite reports on mTEC[low] heterogeneity from RNA-sequencing data[15,16], attempts to study their functional specialisation are limited by a current inability to identify, isolate and study individual mTEC[low] subsets.

To address this, we examined mTEC[low] in the 8–12-week adult mouse thymus. By screening disaggregated TEC-enriched cell suspensions for reactivity with antibodies to surface, secreted and intracellular factors, we identified two distinct mTEC[low] subsets distinguished by their differential cell surface expression of the integrin β4 (CD104) (Fig. 1a), shown previously to be expressed by TECs within the mTEC compartment[16]. Importantly, further analysis showed that CD104[+] mTEC[low], but not CD104[−] mTEC[low], could also be defined by their intracellular expression of CCL21 (Fig. 1a), a chemokine essential for the recruitment of positively selected thymocytes into the medulla[17]. Thus, CD104

expression in mTEC[low] defines a functionally important CCL21-expressing mTEC[low] subset that can be readily isolated by cell sorting. Furthermore, we found that thymic tuft cells[15,16], defined here by the expression of the tuft cell-specific enzyme DCLK1[18], were detectable specifically within the CD104[−]CCL21[−] mTEC[low] subset (Fig. 1a). A CD104[−]CCL21[−]DCLK1[−] 'triple-negative' mTEC[low] subset was also detectable (Fig. 1a). Interestingly, flow cytometric analysis showed that thymic tuft cells were markedly absent from the CD104[+]CCL21[+] mTEC[low] subset (Fig. 1a). In agreement with this, we detected the expression of the tuft cell signature genes, Pou2f3, Dclk1 and Trpm5[19,20], in FACS (fluorescence-activated cell sorting)-sorted CD104[−], but not in CD104[+] mTEC[low] (Fig. 1b). Furthermore, immunofluorescence confocal analysis of adult thymus tissue sections from CCL21[tdTOM] reporter mice confirmed the non-overlapping nature of DCLK1[+] thymic tuft cells and CCL21[+] TEC subsets, and showed individual tuft cells embedded within a network of CCL21[+] mTECs (Fig. 1c). Finally, comparison of the ontogenetic appearance and frequency analysis of mTEC[low] subsets identified CD104[+]CCL21[+] cells as the dominant mTEC[low] population from birth onwards, followed by CD104[−]CCL21[−]DCLK1[−] cells, with the frequency of both populations remaining constant in this period (Fig. 1d). In contrast, and in agreement with other studies[15,16], we found that thymic tuft cells were barely detectable in the neonatal thymus, but then increased in proportion and number during adulthood (Fig. 1d). Collectively, these data define discrete populations within the mTEC[low] compartment of adult mice, and offers opportunities for their isolation and examination of their functional properties.

**LTβR is a critical regulator of multiple mTEC[low] subsets.** To assess the developmental requirements and functional properties of the mTEC subsets described above, we analysed their expression of LTβR, a known regulator of the mTEC lineage[21–23]. All mTEC[low] subsets expressed LTβR, with the highest levels detectable on thymic tuft cells (Supplementary Fig. 1). To examine the potential importance of LTβR expression in the control of mTEC[low] development, we generated Foxn1[Cre]LTβR[fl/fl] 'LTβR[TEC]' mice, where the absence of LTβR in the thymus is selective to the TEC compartment[22]. In particular, given the ability to use a combination of DCLK1/CD104/CCL21 expression to positively identify both thymic tuft cells and CD104[+]CCL21[+] mTECs, we focussed our attention on these two mTEC[low] subsets. Strikingly, in both flow cytometric (Fig. 2a) and confocal analysis (Fig. 2b), we found that thymic tuft cells were absent from the thymus of LTβR[TEC] mice. In agreement with this importance of LTβR, tuft cells were also absent from the thymus of germline Ltbr[−/−] mice (Supplementary Fig. 2). Consistent with the absence of tuft cells in LTβR[TEC] thymus, messenger RNA (mRNA) expression of tuft cell genes, Pou2f3, Dclk1 and Trpm5, was absent from CD104[−] mTEC[low] isolated from LTβR[TEC] mice (Fig. 2c). Collectively, these findings indicate that TEC expression of LTβR is important for thymic tuft cell development. However, as LTβR is expressed by multiple mTEC populations[22], including tuft cells (Supplementary Fig. 1a), it is not clear where in the mTEC developmental pathway LTβR is required for tuft cell development. As a first step to further understand how and when LTβR is important for thymus tuft cell generation, we stimulated alymphoid 2-deoxyguanosine (2dGuo)-treated foetal thymic organ culture (FTOC) for 4 days with an agonistic anti-LTβR antibody[24]. As 2dGuo FTOC lack tuft cells (Fig. 2d) but contain mTEC progenitors[25], this approach provided an initial opportunity to see if LTβR stimulation of mTEC progenitors can induce tuft cell development. As an indicator of the efficacy of antibody treatment of these cultures, anti-LTβR stimulation had a positive

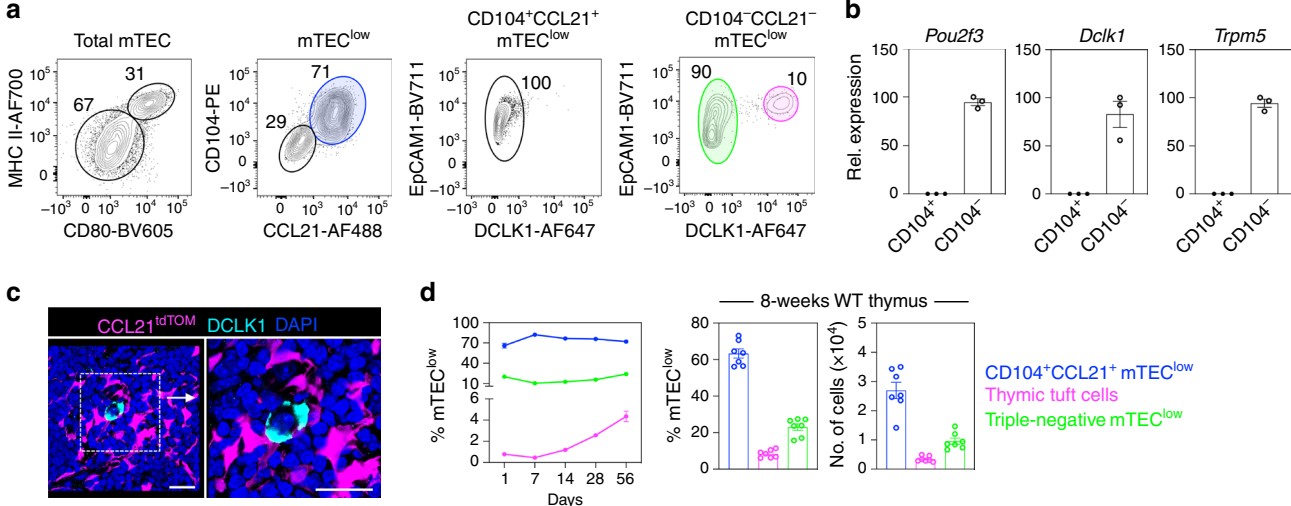

**Fig. 1 The adult mTEC^low compartment is phenotypically diverse. a** EpCAM1+Ly51−UEA1+ total mTECs from digested adult C57/BL6 (WT) thymus were subdivided into mTEC^low and mTEC^hi on the basis of MHCII and CD80 expression. Cell surface (CD104) and intracellular (DCLK1 and CCL21) expression within total mTEC^low is shown, which identifies three mTEC^low subsets: CD104+CCL21+ cells (blue), CD104−CCL21−DCLK1+ thymic tuft cells (magenta) and CD104−CCL21−DCLK1− 'triple-negative' cells (green). **b** qPCR analysis showing mRNA levels of the tuft cell signature genes *Pou2f3*, *Dclk1* and *Trpm5* in CD104+ and CD104− mTEC^low subsets that were FACS sorted from WT adult thymus, data are representative of three biological sorts. **c** Confocal microscopy of frozen tissue sections of adult CCL21^tdTOM thymus stained with antibodies to DCLK1 (cyan), representative of n = 3 independent biological samples. Counterstaining with DAPI is in blue, CCL21^tdTOM is in magenta and scale bars denote 20 μm. **d** Analysis of the ontogenetic frequency of mTEC^low subsets identified in **a**, WT mice of 1 day, 1 week, 2 week, 4 week and 8 week were analysed, CD104+CCL21+ cells: 1 day n = 10, 1 week n = 10, 2 week n = 7, 4 week n = 7, 8 week n = 6, tuft cells: 1 day n = 9, 1 week n = 9, 2 week n = 6, 4 week n = 7, 8 week n = 10, triple-negative cells: 1 day n = 6, 1 week n = 6, 2 week n = 6, 4 week n = 6, 8 week n = 7, over three independent experiments per age. All data are represented as mean ± SEM. Source data are provided as a Source Data file.

effect in FTOC, as indicated by strong upregulation of *Ccl21a* mRNA (Fig. 2e). However, we found that anti-LTβR stimulation did not induce the appearance of thymic tuft cells, as indicated by the absence of DCLK1+ cells by flow cytometry (Fig. 2d), and the absence of *Pou2f3* and *Dclk1* mRNA by quantitative polymerase chain reaction (qPCR) (Fig. 2e). Collectively, these findings demonstrate that while LTβR is an important regulator of thymic tuft cell development, LTβR stimulation of 2dGuo FTOC that contain mTEC progenitors is not sufficient for their development.

We next examined how LTβR expression might influence the development of other mTEC^low subsets. Of particular relevance here, while germline *Ltbr^−/−* mice have a reported reduction in CCL21+ mTEC^low[26], the absence of LTβR expression from all cell types due to germline deficiency did not allow for discrimination between TEC-intrinsic and TEC-extrinsic roles for LTβR in mTEC regulation. To address this, we examined CD104+CCL21+ mTEC^low in LTβR^TEC mice, where in the thymus LTβR is selectively absent from TEC. While the percentage of these cells within the total adult mTEC^low compartment was comparable between Foxn1^Cre controls and LTβR^TEC mice, their absolute numbers were reduced in the latter (Fig. 3a). Interestingly, despite this numerical difference, levels of both CCL21 protein (Fig. 3b) and mRNA (Fig. 3c) were comparable in CD104+CCL21+ mTEC^low, which were isolated from LTβR^TEC mice and Foxn1^Cre controls. Thus, while LTβR may not be an absolute requirement for the developmental emergence of CD104+CCL21+ mTEC^low, including their expression of the chemokine CCL21, it represents an important regulator of the intrathymic availability of these cells. Consistent with this, and the induction of *Ccl21a* mRNA (Fig. 2f), stimulation of 2dGuo FTOC with agonistic anti-LTβR caused a significant increase in the number of CD104+CCL21+ mTEC^low (Fig. 3d). Finally, given that LTβR^TEC mice show combined deficiencies in both thymic tuft cells and CD104+ CCL21+ mTEC^low, we wondered whether the reduction in

CD104+CCL21+ mTEC^low in LTβR^TEC mice may be a consequence of the absence of DCLK1+ thymic tuft cells. To address this, we examined the mTEC^low compartment of *Pou2f3^−/−* mice that lack tuft cells[27]. Interestingly, while thymic tuft cells were absent from these mice as expected, CD104+CCL21+ mTEC^low were present in normal numbers (Supplementary Fig. 3). Collectively, these data identify LTβR as an important regulator of the mTEC lineage through its control of cellular heterogeneity within the mTEC^low compartment.

**LTβR in thymic epithelium controls intrathymic iNKT cells**. In the adult thymus, mTECs play well-defined roles in conventional αβT cell development[4,17]. However, their influence on other functions of the thymus medulla is poorly understood. In particular, it is not clear whether the requirement for mTECs in the development of conventional αβT cells is similar to, or distinct from, their importance for non-conventional T cell development[28]. To study this, we examined how LTβR-mediated control of mTEC diversity might affect the intrathymic development of CD1d-restricted iNKT cells that depend upon medullary microenvironments[28]. We used flow cytometry and PBS57/CD1d tetramers to specifically identify iNKT cells in thymocyte suspensions from adult LTβR^TEC and Foxn1^Cre control mice. Using this approach, we saw a striking reduction in the percentage and absolute number of iNKT cells in the thymus of LTβR^TEC mice (Fig. 4a, b). Next, we subdivided total PBS57/ CD1d+ iNKT cells into distinct subsets on the basis of expression of the transcription factors PLZF, Tbet and RORγt, to identify Tbet+ NKT1, PLZF^hi NKT2 and RORγt+ NKT17 sublineages[7,8,29,30]. Interestingly, we saw that the reduction in total intrathymic iNKT cells was a result of an equivalent (~3–4-fold) reduction in all three iNKT sublineages (Fig. 4b). Importantly, this reduction in iNKT cell numbers in the thymus of LTβR^TEC mice was not due to a reduction in the numbers of

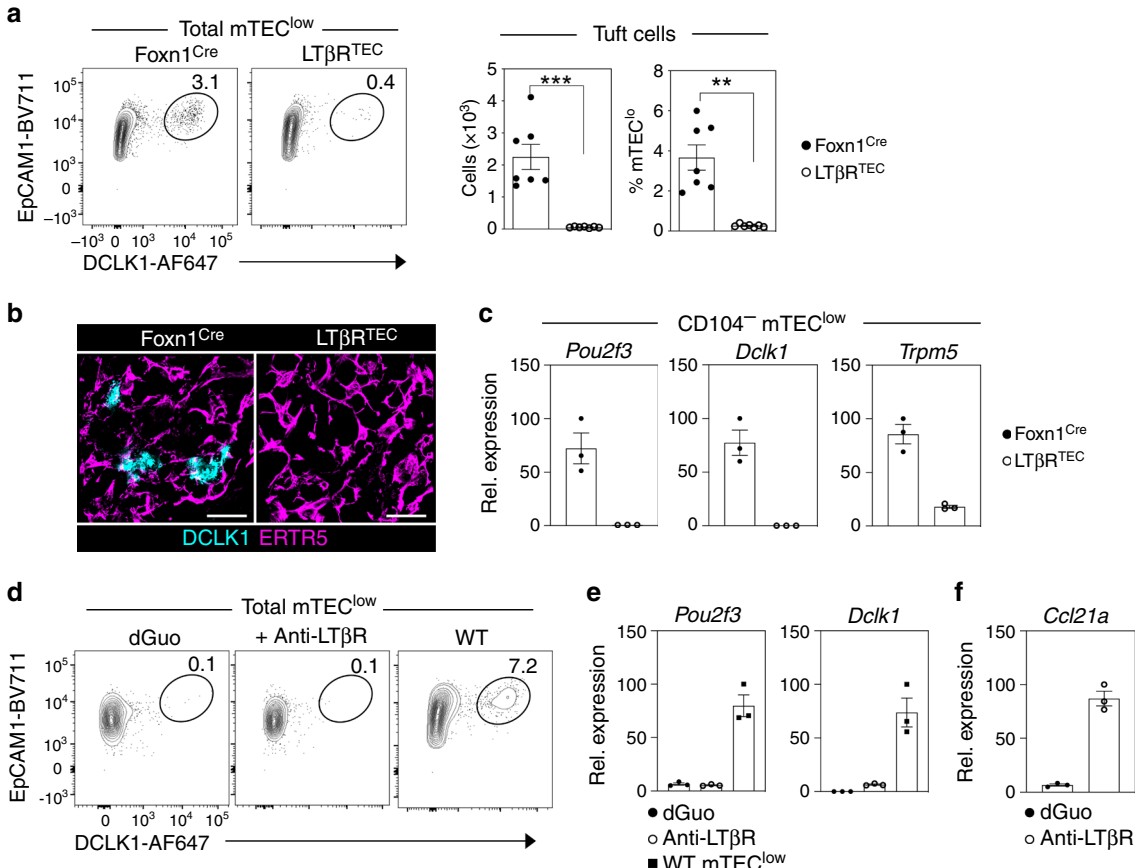

**Fig. 2 LTβR regulates thymic tuft cell development. a** Intracellular staining of mTEC[low] from control Foxn1[Cre] and LTβR[TEC] mice for expression of the tuft cell marker DCLK1. Bar graphs indicate absolute cell numbers and percentages within mTEC[low] in Foxn1[Cre] mice (closed symbols) and LTβR[TEC] mice (open symbols) n = 7 biologically independent samples, over three independent experiments. Significant P values using two-tailed unpaired t test as follows: no. of tuft cells p = 0.0001, % tuft cells p = 0.002. **b** Confocal image of thymus section from Foxn1[Cre] and LTβR[TEC] mice, stained for the mTEC marker ERTR5 (magenta) and DCLK1 (cyan), representative of n = 3 independent biological samples. Scale bars denote 20 μm. **c** qPCR analysis of CD104− mTEC[low] FACS sorted from Foxn1[Cre] (closed symbols) and LTβR[TEC] (open symbols) mice, for expression of the tuft cell markers Pou2f3, Dclk1 and Trpm5, data are representative of three biological sorts. **d** Alymphoid 2dGuo-treated FTOC cultured for 4 days in the presence or absence of agonistic anti-LTβR (2 μg/ml) were analysed by flow cytometry for intracellular expression of DCLK1 to detect tuft cells. Tuft cells in freshly isolated adult WT mTEC[low] are shown for comparison, data are representative of three independent experiments. **e** qPCR analysis of control (closed circles) and anti-LTβR stimulated (open circles) dGuo FTOC for expression of Pou2f3 and Dclk1, with levels of mRNA expression in adult mTEC[low] (closed squares) for comparison, data are representative of three biological sorts from six FTOC cultured lobes/sort. **f** qPCR analysis of control (closed cirlces) and anti-LTβR stimulated (open circles) dGuo FTOC for expression of Ccl21a. All data are represented as mean ± SEM. ***P < 0.001 and **P < 0.01. Source data are provided as a Source Data file.

CD4+CD8+ thymocytes that give rise to iNKT cells (Supplementary Fig. 4). Moreover, and in agreement with previous studies demonstrating the predominant medulla localisation of iNKT cells in wild-type (WT) mice[8], iNKT cells in both Foxn1[Cre] controls and LTβR[TEC] mice were within medullary thymic areas (Supplementary Fig. 4), arguing against the idea that the observed iNKT cell defects are caused by failure to enter the thymus medulla.

To examine how LTβR-mediated control of mTEC[low] might impact iNKT cell development, we examined both cell types for their expression of corresponding receptor/ligand pairs that might explain the iNKT cell defect in LTβR[TEC] mice. First, we analysed cytokine receptor expression by NKT1, NKT2 and NKT17 subsets using flow cytometry. We found IL-17RB, a receptor for the cytokine IL-25 that increases splenic iNKT cell numbers in vitro[31,32], was expressed by both thymic NKT2 and NKT17, but not NKT1 (Fig. 4c). By contrast, the expression of CD122, a component of the receptor for IL-15 that drives iNKT cell proliferation[7,33,34], was expressed by all iNKT sublineages, with the highest expression in NKT1 (Fig. 4d). Next, we analysed

mTEC subsets for expression of corresponding cytokines that interact with the receptors described above. Consistent with the presence of thymic tuft cells in this population, we found CD104− mTEC[low] isolated from WT mice expressed the tuft cell-specific cytokine IL-25 (Fig. 4e). Moreover, CD104− mTEC[low] isolated from LTβR[TEC] mice did not express Il25 mRNA (Fig. 4f), a finding in agreement with the lack of thymic tuft cells in these mice (Fig. 2). In contrast, compared to CD104− mTEC[low], mRNA levels of Il15 and Il15ra that collectively facilitate IL-15 transpresentation[29,30] were highest in CD104+ mTEC[low] (Fig. 4e). Interestingly, Il15 and Il15ra mRNA levels in CD104+ mTEC[low] from LTβR[TEC] mice were comparable to that seen in Foxn1[Cre] controls (Fig. 4f). Thus, within mTEC, IL-25 and IL-15 transpresentation are controlled by LTβR in different ways. First, IL-25 expression in the thymus is LTβR dependent as a consequence of its essential role in thymic tuft cell development. Second, LTβR can influence IL-15 transpresentation in the thymus not by direct control of Il15/Il15ra mRNA expression, but by controlling the frequency of CD104+CCL21+ mTEC[low] that express these genes.

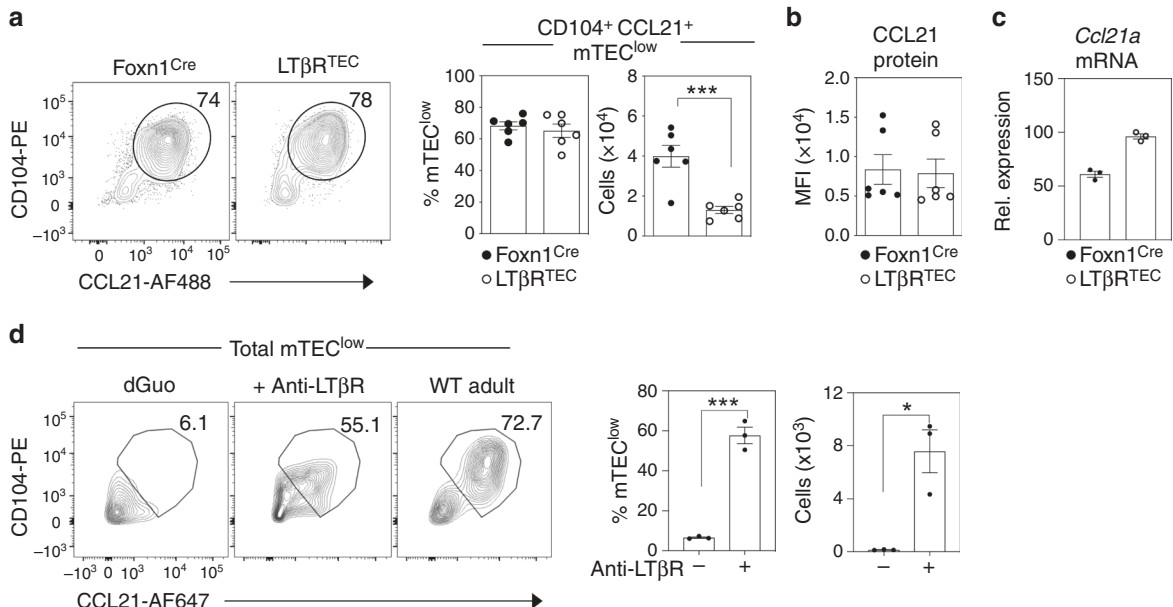

**Fig. 3 Impaired production Of CD104⁺CCL21⁺ mTEC^low In LTβR^TEC mice. a** mTEC^low from Foxn1^Cre and LTβR^TEC mice, analysed for surface expression of CD104 and intracellular expression of CCL21. Bar graphs show cell percentages and absolute cell numbers in Foxn1^Cre (n = 6 biologically independent samples, closed symbols) and LTβR^TEC mice (n = 6 biologically independent samples, open symbols), over three independent experiments. Significant P values using two-tailed unpaired t test are as follows: no. of cells P = 0.0009. **b** MFI levels of intracellular CCL21 expression in pre-gated CD104⁺CCL21⁺ mTEC^low from Foxn1^Cre (closed symbols) and LTβR^TEC (open symbols) mice, n = 6 biologically independent samples, over three independent experiments. **c** Levels of *Ccl21* mRNA in FACS sorted CD104⁺ mTEC^low from Foxn1^Cre (closed symbols) and LTβR^TEC mice (open symbols), data representative of three biological sorts. **d** Alymphoid 2dGuo-treated FTOC cultured for 4 days in the presence or absence of agonistic anti-LTβR (2 μg/ml) were pooled and analysed by flow cytometry for the expression of CCL21 and CD104. Freshly isolated adult WT mTEC^low were stained alongside for comparison. Significant P values using two-tailed unpaired t test are as follows: % cells P = 0.0003, no. of cells P = 0.0102. Six FTOC cultured lobes were pooled per data point, n = 3 biologically independent samples, over three independent experiments. All data are represented as mean ± SEM. *P < 0.05 and ***P < 0.001. Source data are provided as a Source Data file.

**Specialisation of mTEC^low influences iNKT cell sublineages.** Next, we investigated the importance of individual mTEC^low subsets and their products on the intrathymic development of specific iNKT subsets. First, we analysed NKT1, NKT2 and NKT17 sublineages in the thymuses of adult WT and tuft cell-deficient *Pou2f3⁻/⁻* mice. Relevant to this, a previous study[15] showed a reduction in NKT1, NKT2 and NKT17 in the thymus of *Pou2f3⁻/⁻* mice. Consistent with this earlier study, we found a reduction in NKT2 (Fig. 5a, b), but in contrast to that reported by Miller et al.[15], we did not find alterations in NKT1 or NKT17 (Fig. 5a, b). Importantly, while both studies demonstrate a role for tuft cells in NKT2 development in the thymus, the mechanism is not known. As tuft cells are the exclusive producers of IL-25 in the thymus[15,16], and as both NKT2 and NKT17 express high levels of IL-25 receptor (IL-17RB) (Fig. 4), we wondered whether this requirement for tuft cells could be explained by their IL-25 production, by examining thymic iNKT cell development in *Il25⁻/⁻* mice. Here, it is important to note that we used Balb/C WT and *Il25⁻/⁻* mice on a Balb/c background, which as reported previously[7] results in skewing of iNKT cells towards NKT2. Importantly, while NKT1 and NKT17 were unaltered compared to WT controls, we found a significant and selective NKT2 decrease in the thymus of *Il25⁻/⁻* mice. This fits well with previous observations demonstrating a selective NKT2 cell reduction in IL-17RB-deficient mice[35]. Taken together, these findings suggest that at least one of the requirements for tuft cells in thymic NKT2 development can be explained by their provision of IL-25.

Given this reduction in NKT2 in *Il25⁻/⁻* mice, we also investigated whether diminished numbers of NKT2 in LTβR^TEC mice might be restored through the addition of IL-25. As all iNKT sublineages are reduced in LTβR^TEC mice (Fig. 4), they

were chosen as the 'rescue' model for these experiments to see whether cytokine administration might restore multiple iNKT subsets in the same thymus, with any effects then being related to the mTEC defects in these mice. Thus, adult LTβR^TEC mice were injected with either phosphate-buffered saline (PBS) or a single dose of IL-25, and we analysed intrathymic iNKT cells 4 days later. Treatment with recombinant IL-25 in vivo caused a significant and selective increase in NKT2 numbers in LTβR^TEC mice, but did not alter numbers of either NKT1 or NKT17, despite robust expression of the IL-25 receptor by the latter (Fig. 5c, d). Interestingly, in vitro treatment of thymocyte suspensions with IL-25 neither increased nor maintained NKT2 cell numbers. Rather, we saw a progressive decline in all iNKT cell sublineages over a 3-day culture period (Fig. 5e). Thus, provision of IL-25 in vivo is sufficient to rescue the NKT2 deficiency in LTβR^TEC mice, but does not promote their rescue in vitro, suggesting that additional intrathymic factors that are not present in suspension also influence NKT2 cells.

While IL-15 transpresentation can increase thymic NKT[28], it is not known whether this is as a result of selective effects on NKT1, NKT2 and/or NKT17. To address this, we injected LTβR^TEC mice with a single dose of IL-15/IL-15Rα complex (Fig. 6a). After 4 days, we saw a significant increase in both thymic NKT1 and NKT17 (Fig. 6b, c), with NKT2 unchanged (Fig. 6b, c). Thus, in vivo provision of IL-15 transpresentation is sufficient to at least partially restore the thymic defect in NKT1 and NKT17 in LTβR^TEC mice. Interestingly, we found that in vitro addition of IL-15/IL-15Rα complexes to thymocyte suspensions from LTβR^TEC mice significantly increased NKT1 (Fig. 6d, e). Moreover, NKT1-treated in vitro with IL-15/IL-5Rα showed increased levels of Bcl2 protein (Fig. 6f), together with an increased

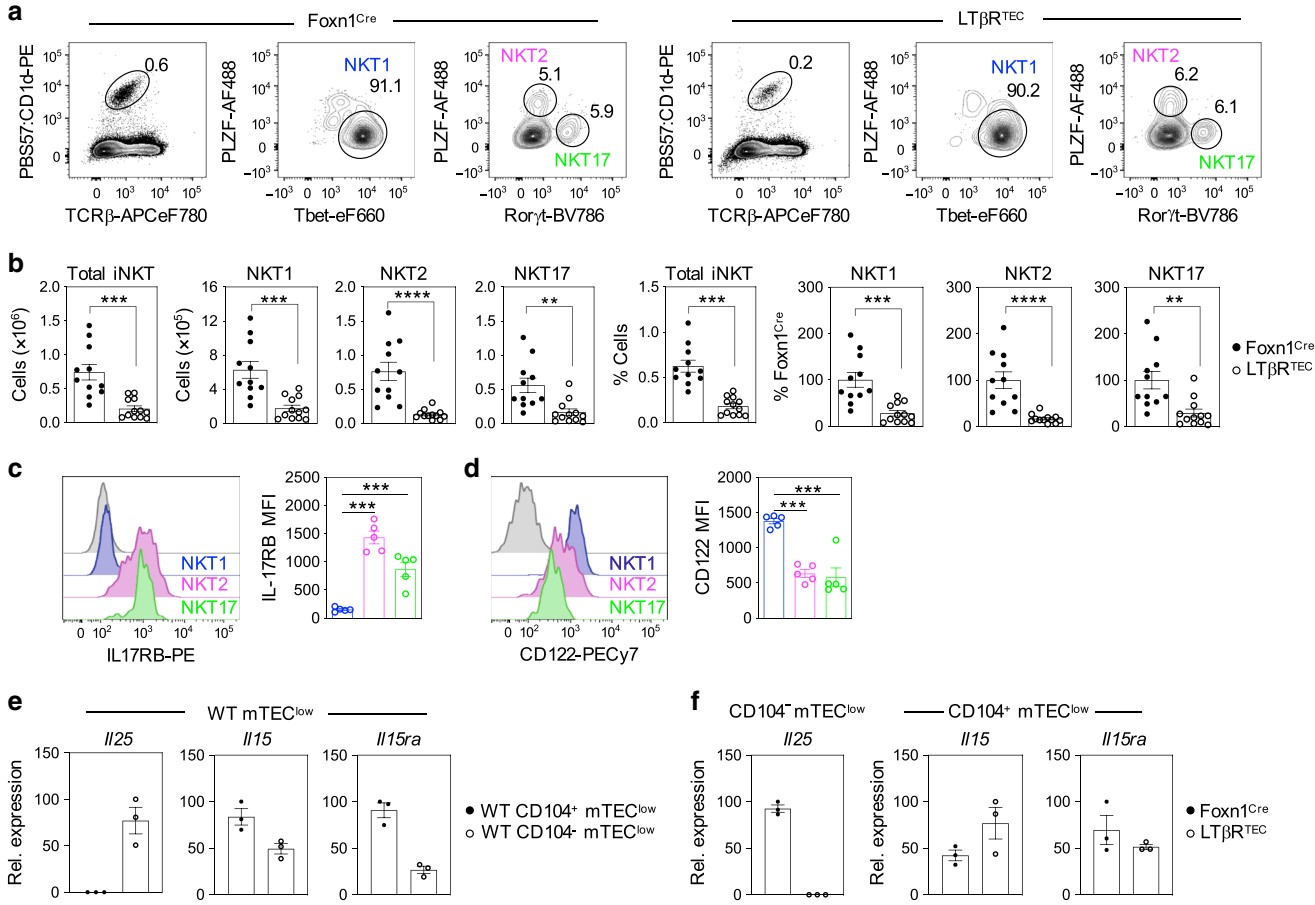

**Fig. 4 LTβR expression by thymic epithelium controls the intrathymic iNKT cell pool. a** Thymocytes from Foxn1[Cre] and LTβR[TEC] mice, analysed for expression of TCRβ and reactivity with the PBS57-loaded CD1d tetramer to identify iNKT cells. TCRβ+PBS57:CD1d+ cells were analysed for intracellular expression of PLZF, Tbet and RORγt to identify NKT1, NKT2 and NKT17 subsets as indicated. **b** Absolute numbers of iNKT cells and iNKT subsets in Foxn1[Cre] controls (n = 11 biologically independent samples, closed symbols) and LTβR[TEC] mice (n = 12 biologically independent samples, open symbols), over three independent experiments. Percentages relative to Foxn1[Cre] controls are also shown. Significant P values using two-tailed unpaired t test as follows: no. of total iNKT P = 0.0002, no. of NKT1 P = 0.0003, no. of NKT2 P ≤ 0.0001, no. of NKT17 P = 0.0021, % Total iNKT P = 0.0001, % NKT1 P = 0.0003, % NKT2 P ≤ 0.0001, % NKT17 P = 0.0021. **c** Levels of expression of IL-17RB (IL-25R) on iNKT subsets; grey histograms represent isotype controls. Bar graphs show MFI of IL-17RB on indicated iNKT subsets, n = 5 biologically independent samples, over three independent experiments. Significant P values using two-tailed unpaired t test as follows: NKT1 vs. NKT2 P ≤ 0.0001, NKT1 vs. NKT17 P = 0.0005. **d** Levels of CD122 expression on iNKT cell subsets; grey histograms represent isotype controls. Bar graphs show MFI of CD122 on indicated iNKT subsets, n = 5 biologically independent samples, over three independent experiments. Significant P values using two-tailed unpaired t test as follows: NKT1 vs. NKT2 P ≤ 0.0001, NKT1 vs. NKT17 P = 0.0004. **e** qPCR analysis of Il25, Il15 and Il15ra mRNA in CD104+ (closed symbols) and CD104− (open symbols) mTEC[low] isolated from adult WT mice. **f** qPCR analysis of Il25 in CD104− mTEC[low], and expression of Il15 and IL15ra in CD104+ mTEC[low] from control Foxn1[Cre] (closed symbols) and LTβR[TEC] mice (open symbols). All data are represented as mean ± SEM, **P < 0.01, ***P < 0.001 and ****P < 0.0001. qPCR was performed in replicate, and data shown are representative of three independent biological sorted samples. Source data are provided as a Source Data file.

frequency of Ki67+ cells (Fig. 6g), suggesting that IL-15 transpresentation regulates NKT1 by controlling their survival and proliferation. Interestingly, and in contrast to in vivo effects, in vitro IL-15/IL-15R treatment did not increase NKT17, suggesting that additional intrathymic factors may also be required alongside IL-15 transpresentation that regulate NKT17 numbers.

**mTEC[low] control iNKT cell function and availability**. The data above suggest that iNKT cell availability in the thymus can be controlled by subset specialisation in mTEC[low], which in turn is regulated by TEC expression of LTβR. We speculated that this axis might represent an important mechanism that then controls the functioning of iNKT cells within multiple tissues. To examine this, we looked in both the thymus and peripheral tissues of LTβR[TEC] mice for evidence of alterations in iNKT cell function.

In the thymus, we focussed attention on DCs as IL-4, a cytokine produced constitutively by thymic iNKT cells, is a known regulator of chemokine production by thymic DCs[7]. We performed flow cytometric analysis of digested thymus preparations to identify CD11c+PDCA1− conventional DCs (cDCs) within the intrathymic DC pool, which we then separated into Sirpα− (cDC1) and Sirpα+ (cDC2) subsets[36,37]. Consistent with our previous study[22], we found that while total numbers of thymic DCs were not significantly altered in LTβR[TEC] mice, there was a significant increase in the number of cDC2. Interestingly, however, we found that the activation status of both cDC1 and cDC2 was diminished in the thymus of LTβR[TEC] mice, as indicated by a reduction in the frequency of MHCII[hi]CD86[hi] cells (Fig. 7a). To correlate these differences in DC activation to the defects in iNKT cells in LTβR[TEC] mice, we next analysed intrathymic DC in Cd1d[−/−] mice. Importantly, these mice have been shown to lack

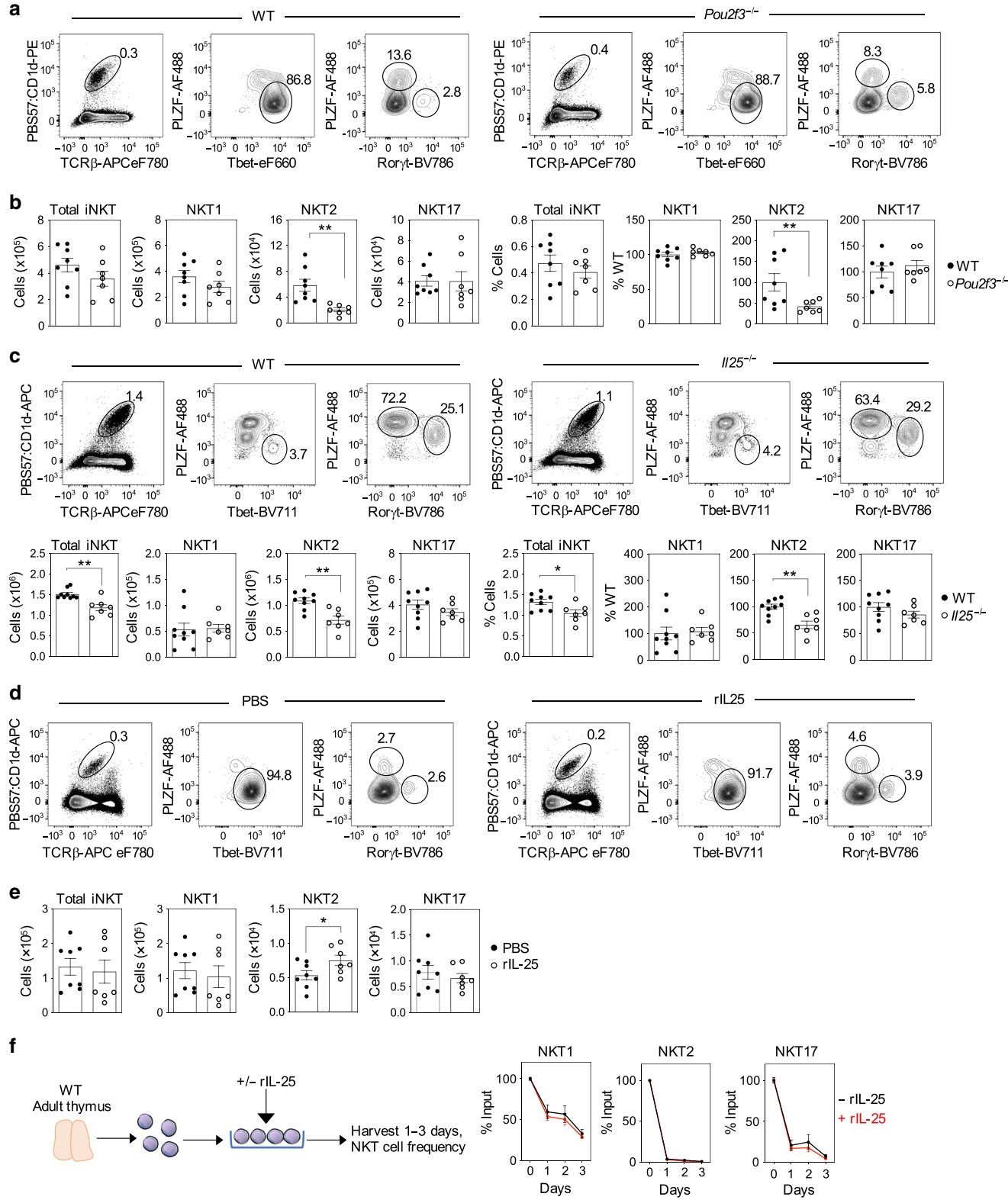

iNKT cells and have been used extensively as a model to study mechanisms that control iNKT cell development in the thymus[7,28]. In these experiments, to avoid any potential confounding problems caused by genetic haploinsufficiency, we used WT mice as controls rather than *Cd1d1* heterozygotes. When we analysed DC activation in *Cd1d1*[−/−] mice, we found that it was reduced in a manner comparable to that seen in LTβR[TEC] mice (Fig. 7b). Thus, these findings further emphasise the regulation of

thymic DC by iNKT cells, and suggest that DC deficiencies in LTβR[TEC] mice may be explained by their deficiencies in iNKT cells.

Perhaps, most significantly, we saw that LTβR[TEC] mice had significant deficiencies in iNKT cells in all peripheral tissues examined, including spleen, lung, liver and peripheral blood (Fig. 8a, b). Thus, despite selective targeting of LTβR deletion by Foxn1[Cre], which targets only the thymus and skin[38], defects in

**Fig. 5 IL-25 production from thymic tuft cells controls intrathymic iNKT2 cells.** Flow cytometric detection (**a**) and quantitation (**b**) of TCRβ+PBS57:CD1d tetramer+ iNKT cells and NKT1, 2 and 17 subsets in WT ($n = 8$ biologically independent samples, closed symbols) and tuft cell-deficient $Pou2f3^{-/-}$ ($n = 7$ biologically independent samples, open symbols), over three independent experiments. Significant $P$ values using two-tailed unpaired $t$ test as follows: no. of NKT2 $P = 0.0033$ and % NKT2 $P = 0.0253$. Flow cytometric detection and quantitation (**c**) of total iNKT and iNKT subsets in BALB/c WT ($n = 9$ biologically independent samples, closed symbols) and $Il25^{-/-}$ ($n = 7$ biologically independent samples, open symbols) mice, over four independent experiments. Significant $P$ values using two-tailed unpaired $t$ test as follows: no. of total iNKT $P = 0.0013$, no. of NKT2 $P = 0011$ and % NKT2 $P = 0.0011$. Flow cytometric detection (**d**) and quantitation (**e**) of total iNKT and iNKT subsets in LTβR^TEC mice, 4 days after injection with either PBS ($n = 8$ biologically independent samples, closed symbols) or recombinant IL-25 ($n = 7$ biologically independent samples, open symbols), over three independent experiments. Significant $P$ values using two-tailed unpaired $t$ test as follows: no. of NKT2 $P = 0.0482$. **f** WT thymocyte suspensions were cultured in the presence (red line) or absence (black line) of recombinant IL-25 for the indicated period, and iNKT subsets were quantitated by flow cytometry. Data are shown as mean percentage of input (i.e. cells at d0), and error bars indicate SEM of triplicate wells from one experiment, representative of three separate experiments. All data are represented as mean ± SEM. *$P < 0.05$ and **$P < 0.01$. Source data are provided as a Source Data file.

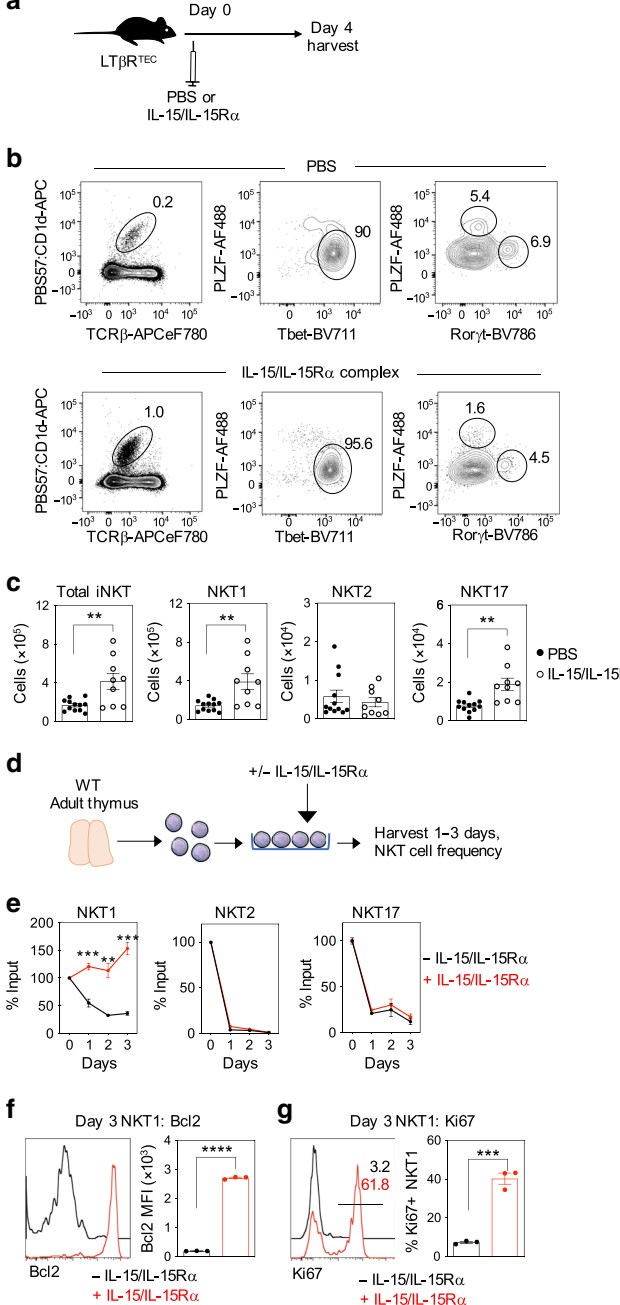

**Fig. 6 IL-15 transpresentation stimulates survival and proliferation of intrathymic iNKT1. a** Schematic showing in vivo treatment of LTβR^TEC mice with PBS or IL-15/IL-15Rα complexes. **b** iNKT populations were analysed by flow cytometry from LTβR^TEC mice injected with PBS (top panel) or IL-15/IL-15Rα complex (bottom panel) 4 days post injection. **c** Quantitation of total iNKT and iNKT subsets in control ($n = 12$ biologically independent samples, closed symbols) and IL-15/IL-15Rα injected mice ($n = 9$ biologically independent samples, open symbols) from five independent experiments. Significant $P$ values using two-tailed unpaired $t$ test as follows: no. of total iNKT $P = 0.0035$, no. of NKT1 $P = 0.0031$, no. of NKT17 $P = 0.0012$. **d** Schematic showing in vitro culture of WT thymocytes in the absence or presence of IL-15/IL-15Rα complexes. **e** WT thymocyte suspensions were cultured in the presence (red line) or absence (black line) of IL-15/IL-15Rα complexes for the indicated period, and iNKT subsets were quantitated by flow cytometry. Data are shown as mean percentage of input (i.e. cells at d0), and error bars indicate SEM of triplicate wells from one experiment, representative of three separate experiments. Significant $P$ values using two-tailed unpaired $t$ test as follows: NKT1 day 1 $P = 0.0008$, NKT1 day 2 $P = 0.0034$, NKT1 day 3 $P = 0.0005$. Analysis of intracellular Bcl2 (**f**) and Ki67 (**g**) expression in untreated (black lines) and IL-15/IL-15Rα-treated (red lines) NKT1 cells at day 3 of culture. Data shown represent a minimum of three separate experiments. Significant $P$ values using two-tailed unpaired $t$ test as follows: Bcl2 MFI $P \leq 0.0001$, % Ki67+ $P = 0.0003$. All data are represented as mean ± SEM. **$P < 0.01$, ***$P < 0.001$ and ****$P < 0.0001$. Source data are provided as a Source Data file.

iNKT cells in the thymus of LTβR^TEC mice are accompanied by long-term iNKT cell defects in extrathymic tissues. Interestingly, splenic iNKT cells in Foxn1^Cre and LTβR^TEC mice responded equally well to in vivo αGal-Cer stimulation in terms of per cell levels of IL-4 and interferon-γ (IFNγ) expression (Fig. 8c, d). In contrast, and in agreement with the reduced iNKT peripheral pool size, fewer IL-4+ and IFNγ+ cells (Fig. 8e), and lower levels of serum IL-4 and IFNγ (Fig. 8f), were detected in LTβR^TEC mice following αGal-Cer stimulation. Thus, despite their reduced frequency in LTβR^TEC mice, iNKT cells are functionally competent in terms of their cytokine production in response to in vivo T cell receptor (TCR) stimulation. Collectively, our data identify an intrathymic mechanism in which LTβR regulates subset specialisation in mTEC, which not only controls iNKT cell development and function in the thymus but also iNKT cell availability in peripheral tissues.

## Discussion

In contrast to the known importance of mTEC^hi in thymic tolerance[4,6,12], functional specialisation in mTEC^low is poorly understood. To address this, we examined the adult mTEC^low compartment and the molecular mediators that control it. Next,

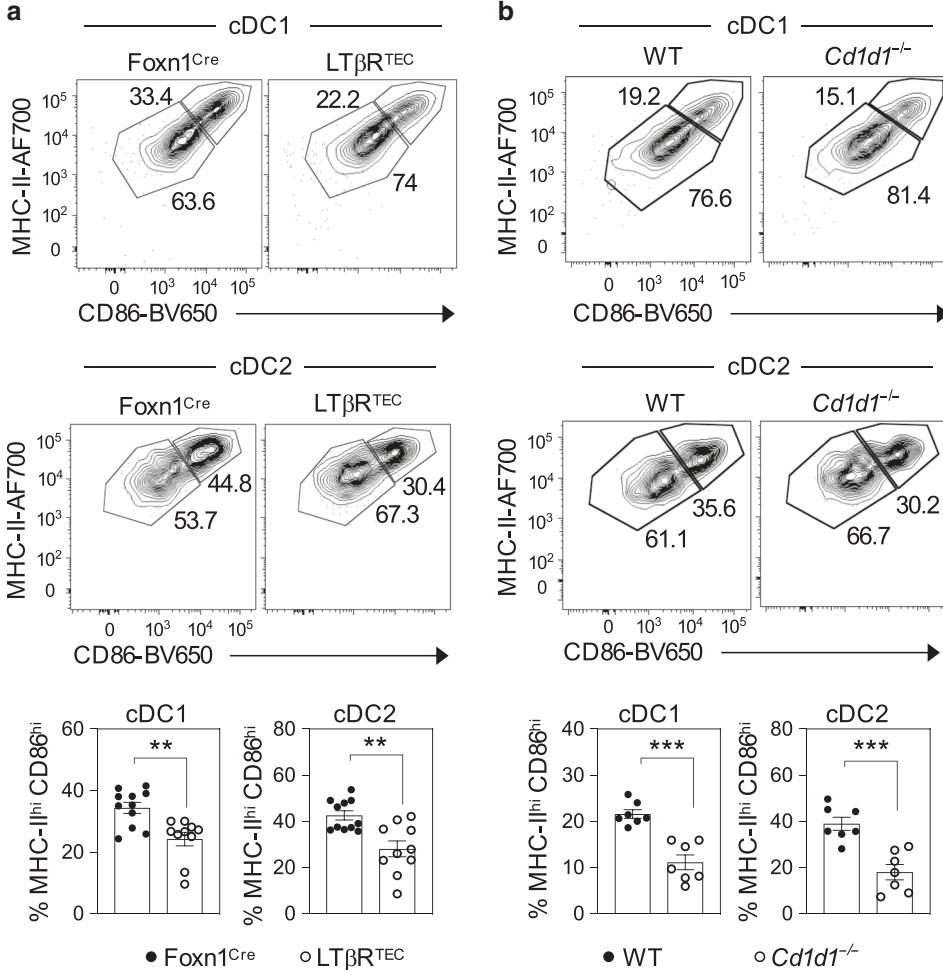

**Fig. 7 Impaired dendritic cell activation in LTβR^TEC mice.** Flow cytometric analysis of CD86 and MHCII expression by Lin⁻CD11c⁺Sirpα⁻ cDC1 and Lin⁻CD11c⁺Sirpα⁺ cDC2 in the thymuses of **a** Foxn1^Cre ($n = 10$ biologically independent samples, closed symbols) and LTβR^TEC ($n = 11$ biologically independent samples, open symbols), and **b** WT ($n = 7$ biologically independent samples, closed symbols) Cd1d1^−/− ($n = 7$ biologically independent samples, open symbols). In both cases, graphs in lower panels show quantitation of MHCII^hiCD86^hi cDC1 and cDC2 (gated as in upper panels). All data are from three independent experiments. Significant P values using two-tailed unpaired t test are as follows: **a** cDC1 $P = 0.0018$, cDC2 $P = 0.0014$, **b** cDC1 $P = 0.0001$, cDC2 $P = 0.0004$. All data are represented as mean ± SEM. *** $P < 0.001$ and ** $P < 0.01$. Source data are provided as a Source Data file.

we analysed the functional significance of mTEC^low heterogeneity during intrathymic iNKT cell development, a known but poorly understood function of the medulla.

Within adult mTEC^low, we identified CD104⁺ cells uniformly expressing the chemokine CCL21, as well as an additional and distinct DCLK1⁺ subset. The latter was specifically contained within CD104⁻ cells and represents recently described thymic tuft cells[15,16]. Thus, and in agreement with single-cell RNA-sequencing data[15,16], mTEC^low are phenotypically heterogeneous. To identify regulators of mTEC^low heterogeneity, we created LTβR^TEC mice where the mTEC regulator LTβR was selectively absent from TECs. Analysis showed that LTβR is essential for thymic tuft cell differentiation, but is not an absolute requirement for CD104⁺CCL21⁺ mTEC^low differentiation. Instead, LTβR influences the frequency of these cells. This reduction in CCL21⁺ mTEC^low in LTβR^TEC mice fits well with a similar observation in germline Ltbr^−/− mice[26]. Significantly, as the absence of LTβR expression in the thymus of LTβR^TEC mice is selective to TECs, our findings extend these observations by demonstrating that LTβR regulates CCL21⁺ mTEC^low via a direct requirement for LTβR expression by TECs. While the mechanism by which LTβR regulates CCL21⁺ mTEC^low is unclear, it may reflect the importance of LTβR in controlling their proliferation and/or cell

turnover[21,26]. Alternatively, a requirement for LTβR by mTEC progenitors[39] may also impact the frequency of CD104⁺CCL21⁺ mTECs. Similarly, although our data shows an important requirement for LTβR in thymic tuft cell development, the stage (s) in the mTEC developmental pathway where LTβR is required are not known. For example, tuft cell deficiency may be caused by a requirement for LTβR in the development of immature mTEC^lo progenitors that represent progenitors of mTEC^hi. Alternatively, as thymic tuft cells can arise from mTEC^hi [15], defects in the maturation of mTEC^hi towards more mature mTEC^low stages may be responsible for the absence of tuft cells. Perhaps, consistent with this, mTEC^hi are also reduced in LTβR^TEC mice[22], and germline Ltbr^−/− mice that lack LTβR in all tissues also lack terminally differentiated mTECs[40]. Finally, it is also interesting to note that tuft cells themselves express detectable levels of LTβR (Supplementary Fig. 1), so a further possibility is that LTβR directly regulates tuft cells themselves, rather than in upstream stages of the mTEC lineage. Thus, in current models of the mTEC developmental pathway, that is, 'bona fide' mTEC^low progenitors > mTEC^hi > terminally differentiated mTEC^low and tuft cells, it is not currently known where the requirement for LTβR in tuft cell development maps to, and tackling this question will require the generation of new mouse models that enable gene targeting at

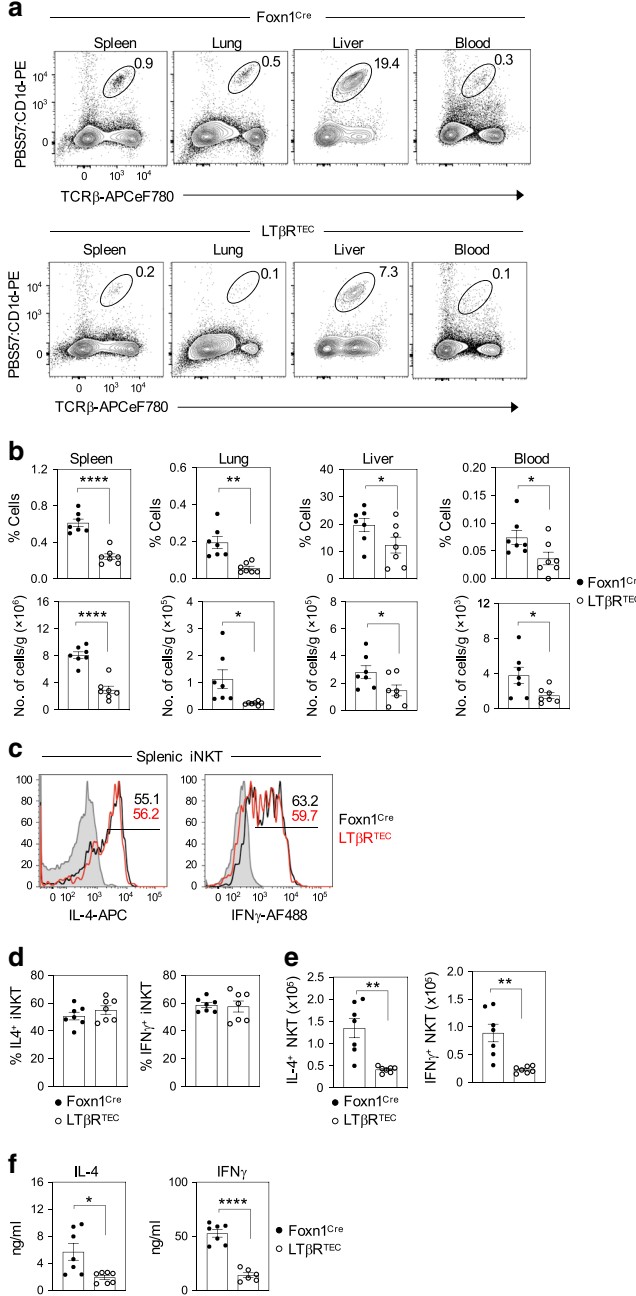

**Fig. 8 LTβR expression by thymic epithelium determines peripheral iNKT-pool size. a** indicated that tissues from control Foxn1^Cre and LTβR^TEC mice were analysed for iNKT cell frequencies by flow cytometry. Graphs in **b** show cell percentages and number of iNKT cells per gram or ml of tissue, as indicated, in Foxn1^Cre ($n = 7$ biologically independent samples, closed symbols) and LTβR^TEC ($n = 7$ biologically independent samples, open symbols), over three independent experiments. Significant $P$ values using two-tailed unpaired $t$ test are as follows: % spleen $P \leq 0.0001$, % lung $P = 0.0012$, % liver $P = 0.0494$, % blood $P = 0.0383$, no. of cells in the spleen $P \leq 0.0001$, no. of cells in the lung $P = 0.0255$, no. cells in the liver $P = 0.0459$, no. of cells in blood $P = 0.0371$. **c** Flow cytometric analysis of intracellular IL-4 or IFNγ expression in splenic iNKT cells in Foxn1^Cre (black lines) and LTβR^TEC (red lines) mice, 2 h after i.v. administration of αGal-Cer. Grey histograms show levels of staining in non-injected mice. Percentages (**d**) and numbers (**e**) of IL-4^+ and IFNγ^+ splenic iNKT after αGal-Cer stimulation of Foxn1^Cre ($n = 7$ biologically independent samples, closed symbols) and LTβR^TEC mice ($n = 8$ biologically independent samples, open symbols), over three independent experiments. Significant $P$ values using two-tailed unpaired $t$ test as follows: no. of IL-4^+ cells $P = 0.0011$, no. of IFNγ^+ cells $P = 0.0014$. **f** shows ELISA quantitation of IL-4 (2 h post Gal-Cer injection) Foxn1^Cre ($n = 7$ biologically independent samples, closed symbols) and LTβR^TEC mice ($n = 7$ biologically independent samples, open symbols), and IFNγ (16 h post Gal-Cer injection) in serum samples of Foxn1^Cre ($n = 7$ biologically independent samples, closed symbols) and LTβR^TEC mice ($n = 6$ biologically independent samples, open symbols), over three independent experiments. Significant $P$ values using two-tailed unpaired $t$ test are as follows: IL-4 $P = 0.0117$, IFNγ $P \leq 0.0001$. All data are represented as mean ± SEM. *$P < 0.05$, **$P < 0.01$, and ****$P < 0.0001$. Source data are provided as a Source Data file.

specific stages of the mTEC lineage. Nevertheless, that LTβR is an essential regulator of thymic tuft cells provides important information on a newly described and poorly understood TEC subset. Interestingly, while thymic tuft cells are absent in both LTβR^TEC and *Ltbr*^−/− germline mice, intestinal tuft cells are still present in *Ltbr*^−/− mice (Supplementary Fig. 5). Thus, despite shared similarities of thymic and intestinal tuft cells, including their production of IL-25[15,41], they are generated via distinct developmental pathways discriminated by a differential requirement for LTβR. As such, these findings demonstrate a tissue-specific role for LTβR in the control of tuft cells, with LTβR^TEC mice providing a model in which tuft cell deficiency is thymus specific.

When we assessed the relevance of the mTEC^low heterogeneity described above to αβT cell development, we saw a striking effect on iNKT cells in LTβR^TEC mice, which was explained by a reduction in all NKT1, NKT2 and NKT17 lineages. This was in contrast to unaltered programmes of conventional and Foxp3^+

regulatory T cell development in LTβR^TEC mice[22]. Thus, LTβR expression by TECs is a selective and important controller of intrathymic iNKT cell development. An interesting feature of the iNKT cell defect in LTβR^TEC mice is that all sublineages show a comparable reduction, suggesting that LTβR does not control lineage specification in iNKT cells. Rather, it may either influence common iNKT cell progenitors[42] or operate at later stages of development by determining how mTECs regulate downstream mature iNKT sublineages. In support of the latter, we found that within the mTEC^low subsets analysed here, CD104^+CCL21^+ mTEC^low expressed the highest levels of both *Il15* and *Il15ra*, genes that are collectively required to enable IL-15 transpresentation. Significantly, in vivo treatment of LTβR^TEC mice with IL-15/IL-15R complexes was sufficient to increase both NKT1 and NKT17, indicating the reduction in these cells in LTβR^TEC mice is likely caused by limited IL-15 transpresentation. Together, these findings suggest that NKT1 and NKT17 can be regulated by IL-15 transpresentation. Importantly, these findings extend earlier observations demonstrating a link between mTECs and iNKT cells, and IL-15 transpresentation[28]. For example, this earlier study did not examine the requirements of individual NKT1, NKT2 and NKT17 subsets, nor did it assess the potential functional importance of mTEC^low subsets described here. Relevant to this, while our study indicates that CD104^+CCL21^+ mTEC^low are capable of IL-15 transpresentation, it is currently unclear whether IL-15 transpresentation is limited to CD104^+ CCL21^+ mTEC^low, or a property shared by other mTEC subsets. Related to this, while analysis of IL-15^CFP knock-in mice demonstrated *Il15* expression in mTEC^hi, expression of *Il15ra* was not analysed in this study[43]. Relevant to this, we showed previously[28] that mTEC^hi express only very low levels of *Il15ra* in comparison to mTEC^low. Thus, while *Il15* expression may be a shared feature of multiple mTEC subsets, differential expression of *IL15Ra* within mTEC may mean that IL-15Rα-mediated

transpresentation is more restricted. Direct analysis of this will require the generation of new mouse models that allow selective gene deletion in specific mTEC subsets. Regardless of how widespread IL-15 transpresentation may be within mTEC, the finding that CCL21+ mTEClow express genes that regulate IL-15 transpresentation extends our current understanding of the involvement of this mTEC subset during T cell development. For example, as well as controlling the CCL21-mediated entry of conventional thymocytes into the medulla following positive selection[17], their regulation of NKT1 and NKT17 by IL-15 transpresentation highlights an additional functional property of CD104+CCL21+ mTEClow and suggest that they act as regulators of multiple αβT cell lineages in the thymus.

In analysing the intrathymic requirements of NKT2 cells, we found that they expressed high levels of IL-17RB, a receptor for the tuft cell-specific cytokine IL-25. Moreover, IL-25 treatment of LTβRTEC mice increased NKT2 numbers in vivo. This correlated with a selective NKT2 defect in tuft cell-deficient Pou2f3−/− mice, where the intrathymic source of IL-25 is absent. That NKT2 cells are reduced in the thymus of tuft cell-deficient Pou2f3−/− mice fits well with a previous similar observation[15]. Importantly, however, our study differs from this earlier report in several ways. First, it is significant that the earlier study did not address the mechanism by which tuft cells influence NKT2 cells. Here, we show that as with Pou2f3−/− mice, iNKT2 are also reduced in the thymus of Il25−/− mice. As thymic expression of IL-25 is exclusive to tuft cells[15,16] our data and that of others collectively provide evidence for a mechanism in which tuft cells control NKT2 cells via their IL-25 production. This scenario fits well with the intrathymic restriction of IL-25 to tuft cells reported by both Miller et al.[15] and Bornstein et al.[16], and is further supported by experiments here showing increased NKT2 cells following in vivo IL-25 administration. Second, and in contrast to Miller et al.[15], we found that NKT1 and NKT17 cells are not significantly reduced in the thymus of Pou2f3−/− mice. The reasons for the discrepancy in these findings on NKT1 and NKT17 cells are unknown. However, as NKT1 cells lack IL-25 receptor expression, any influence of tuft cells is unlikely to be explained by a requirement for the tuft cell-specific product IL-25. In support of this, we found that NKT1 cells were not significantly altered in Il25−/− mice. Interestingly, while NKT17 cells do express dectectable levels of IL-25 receptor, our analysis showing unaltered numbers of NKT17 cells in tuft cell-deficient mice was paralleled by normal numbers of NKT17 cells in IL-25 -eficient mice. Thus, in our hands, and unlike for NKT2 cells, tuft cells and IL-25 do not appear to be essential regulators of either NKT1 or NKT17 cells. Finally, despite our evidence indicating a role for IL-25 in regulating NKT2 cells in vivo, we found that IL-25 did not increase NKT2 cells in vitro. Together, these findings suggest that through the production of IL-25, tuft cells represent an important and selective regulator of NKT2 cells, and that this cytokine operates in synergy with other intrathymic signals to control their numbers. While the nature of these signals is not known, it is interesting that IL-25 stimulates proliferation of splenic iNKT cells in the presence of TCR stimulation[31]. As thymic iNKT cells show evidence of ongoing TCR signalling[7,9], and bone marrow-derived thymic antigen-presenting cell express the iNKT cell receptor ligand CD1d[9], this raises the possibility that tuft cell production of IL-25 operates in conjunction with TCR stimulation in the thymus medulla to regulate NKT2 cells. Perhaps, relevant to this, we found evidence for a functional consequence of reduced iNKT cell development in LTβRTEC mice. Here, activation of intrathymic DCs was diminished, which mirrored DC defects in Cd1d1−/− mice. How iNKT cells regulate thymic DC activation is not clear. Indeed, while this study uses Cd1d1−/− mice to analyse this requirement for iNKT cells, future analysis of Ja18−/− mice[44]

that exclusively lack type 1 iNKT cells may help to address this. Whatever the case, this scenario fits well with the ability of cytokine production by NKT2 cells (i.e. IL-4) to regulate the expression of chemokines (CCL17 and CCL22) by thymic DCs[7]. Alongside this impact of the NKT2-produced cytokine IL-4 on thymic DCs, it may be the case that additional cytokines from other iNKT subsets can also influence thymic DCs. For example, it is tempting to speculate that intrathymic production of IFNγ, a signature cytokine expressed by thymic NKT1 cells[7] and known regulator of DC activation[45], may also be an explanation for how iNKT cells regulate thymic DCs.

Finally, a striking finding in LTβRTEC mice was the reduction in iNKT cells in both the thymus and multiple peripheral tissues. These findings differ from previous studies on germline Ltbr−/− mice, which showed reduced NKT cells in the liver and spleen, but normal numbers in the thymus[46,47]. There are several possibilities to explain this discrepancy. First, in germline Ltbr−/− mice peripheral lymphoid organ development is profoundly disrupted, including the absence of lymph nodes and loss of splenic organisation[48], which may alter the survival and/or proliferation of iNKT cells following their exit from the thymus. Second, germline Ltbr−/− mice also have defects in thymus emigration, which results in an intrathymic accumulation of mature thymocytes[21]. Thus, a reduction in thymus emigration could also explain the reduction in iNKT numbers in peripheral tissues of germline Ltbr−/− mice. Consequently, these earlier studies could not determine whether iNKT abnormalities caused by germline LTβR deletion are a result of altered events in the thymus or the periphery, or both. Importantly, in LTβRTEC mice, selective intrathymic targeting of LTβR deletion to TEC with Foxn1Cre directly demonstrates that loss of iNKT cells in these mice is due to defective intrathymic development. Moreover, unaltered thymus emigration in LTβRTEC mice[49] also argues against an impact on iNKT cells caused by defects in this process. Thus, our data indicate that TEC expression of LTβR influences intrathymic development of NKT cells in the thymus, and that this then causes iNKT cell defects in the periphery.

Importantly, the defect in peripheral iNKT cells in LTβRTEC mice is long lived and not accompanied by a recovery in their numbers in peripheral tissues. In line with diminished peripheral iNKT-pool size, we saw reduced levels of serum cytokine production by peripheral iNKT cells following in vivo stimulation of LTβRTEC mice with αGal-Cer. However, IL-4 and IFNγ production by iNKT cells on a per cell basis was normal, suggesting that defects in intrathymic maturation do not impair their response to TCR stimulation. Why iNKT cell numbers do not recover in peripheral tissues following defective thymus development is unclear. One possibility is that undefined LTβR-dependent signals are needed by iNKT cells during intrathymic maturation, and their absence impairs long-term iNKT cell fitness in the periphery. Alternatively, possible alterations in conventional or Foxp3+ T-reg that might be caused by loss of LTβR may place iNKT cells at a competitive disadvantage in peripheral tissues.

In conclusion, our study provides new evidence for the importance of mTEClow heterogeneity in adult mice. Our identification of LTβR as a key regulator of multiple mTEClow subsets, each controlling individual iNKT sublineages, helps redefine our understanding of how the medulla influences non-conventional αβT cells. Finally, that iNKT cell availability in multiple organs is controlled through expression of LTβR by TECs demonstrates the importance of the thymus in the regulation of the peripheral immune system.

## Methods
**Mice**. The following mice were sacrificed via cervical dislocation at 8–12 weeks of age, unless otherwise indicated: WT mice on a C57BL/6 and BALB/c background

were bred at the University of Birmingham, B6 *Pou2f3*[−/−][27] (RIKEN BioResource Research Centre), germline *Ltbr*[−/−][48] (provided by Klaus Pfeffer, Heinrich Heine University, Germany), CCL21[tdTOM][17], B6 *Cd1d1*[−/−][50] (JAX), and BALB/c *Il25*[−/−][51] (provided by Andrew McKenzie, LMB Cambridge). LTβR[TEC] mice were generated by crossing Foxn1[Cre] mice[38] with LTβR[fl/fl] mice[52] (provided by Alexei Tumanov, UT Health San Antonio, Texas). Foxn1[Cre] mice were used as controls for LTβR[TEC] mice. All mice were co-housed in standard barrier conditions at the Biomedical Services Unit at the University of Birmingham, and equal proportions of male and female mice were used in all experiments. All mouse experiments were done with permission from the Birmingham Animal Welfare and Ethical Review Body and the UK Home Office.

**Flow cytometry and cell sorting.** For iNKT analysis, thymus, spleen and liver samples were mechanically disrupted. Lungs were enzymatically digested using Liberase TM (Roche, 12.5 μg/ml) and DNase I (100 mg/ml, Roche), and subsequently treated with Red Cell Lysis Buffer (Sigma). Cell suspensions were stained with the following: anti-TCRβ (H57-597, eBioscience, cat. no.: 47-5961-82, 1:200), anti-PLZF (Mags.21F7, eBioscience, cat. no.: 53-9320-82, 1:100), anti-RORγt (Q31-378, Becton Dickinson, cat. no.: 564723, 1:100), anti-T-bet (4B10, eBioscience, cat. no.: 50-5825-82, 1:100), anti-T-bet (4B10, BioLegend, cat. no.: 644820, 1:100), anti-CD122 (TM-b1, eBioscience, cat. no.: 25-1222-82, 1:100), anti-IL-17RB (MUNC33, eBioscience, cat. no.: 12-7361-82, 1:100) and CD1d tetramers loaded with PBS57 or unloaded CD1d tetramers (obtained from the National Institutes of Health Tetramer Facility). For stromal cell analysis, thymus tissue was digested with collagenase dispase (2.5 mg/ml, Roche) and DNase I (40 mg/ml, Roche) and depleted of CD45[+] cells using anti-CD45 microbeads and LS columns (Miltenyi Biotec). TECs were analysed using the following antibodies: anti-CD45 (30-F11, eBioscience, cat. no.: 47-0451-82, 1:1000), anti-EpCAM1 (G8.8, eBioscience, cat. no.: 46-5791-82, 1:2000), anti-Ly51 (6C3, eBioscience, cat. no.: 17-5891-82, 1:1000), anti-IA/IE (M5/114.15.2, eBioscience, cat. no.: 56-5321-82, 1:2000), anti-CD80 (16-10A1, BioLegend, cat. no.: 104729, 1:800) and anti-CD104 (346-11A, BioLegend, cat. no.: 123610, 1:1000). Biotinylated UEA-1 (Vector Laboratories, B-1065, 1:10,000) was detected using streptavidin PECy-7 (eBioscience, cat. no.: 25-4317-82, 1:1500). Intracellular detection of CCL21 and DCLK1 was achieved using anti-CCL21 (Lifespan Biosciences, cat. no.: LS-C104634, 1:100) and anti-DCLK1 (DCAMKL1, Abcam, cat. no.: Ab31704, 1:1000), followed by donkey anti-rabbit Alexa Fluor 488 (Invitrogen, cat. no.: A-21206, 1:1000). Thymic DCs were analysed following digestion using collagenase D (2.5 mg/ml, Roche) and DNase I (40 mg/ml, Roche), and stained with anti-CD11c (N418, eBioscience, cat. no.: 25-0114-82, 1:800), anti-PDCA-1 (129C1, BioLegend, cat. no.: 127018, 1:200), anti-Sirpα (P84, eBioscience, cat. no.: 12-1721-82, 1:200), anti-IA/IE (M5/114.15.2, eBioscience, cat. no.: 56-5321-82, 1:200) and anti-CD86 (GL1, BioLegend, cat. no.: 105035, 1:200). Anti-CD3 (145-2C11, eBioscience, cat. no.: 47-0031-82, 1:200), anti-CD19 (eBio1D3, eBioscience, cat. no.: 47-0193-82, 1:200) and anti-NK1.1 (PK136, eBioscience, cat. no.: 47-5941-82, 1:200) were used to create a lineage dump gate prior to DC detection. All intracellular staining was performed using a Foxp3 staining buffer set (eBioscience). Cultured thymocytes were analysed for expression of Bcl2 and Ki67 using anti-Bcl2 (BCL/10C4, BioLegend, cat. no.: 635510, 1:100) and anti-Ki67 (SolA15, eBioscience, cat. no.: 25-5698-82, 1:2000). Cells were analysed on an LSR Fortessa (Becton Dickinson), and data processed using the FlowJo software (v.8 and v.10). The following TEC subsets were sorted using a FACS Aria Fusion I (Becton Dickinson): CD104[−]mTEC[low] (CD45[−]EpCAM[+]Ly51[−]UEA1[+]MHC-II[low]CD104[−]), CD104[+]mTEC[low] (CD45[−]EpCAM[+]Ly51[−]UEA1[+]MHC-II[low]CD104[+]) and total mTEC[low] (CD45[−]EpCAM[+]Ly51[−]UEA1[+]MHC-II[low]CD80[low]). Gating strategies for flow cytometry and sorting can be found in Supplementary Fig. 6.

**Confocal microscopy.** Seven micrometre sections were cut from snap-frozen thymus tissue, and then fixed in acetone. Antibodies used were: anti-DCLK1 (DCAMKL1, Abcam, cat. no.: Ab31704, 1:500), anti-CD8 (53-6.7, BioLegend, cat. no.: MCD0821, 1:100), ERTR5 (hybridoma supernatant used undiluted)[53]. Secondary antibodies used were donkey anti-rabbit Alexa Fluor 555 (Invitrogen, cat. no.: A-31572, 1:1000) and goat anti-rat IgM Alexa Fluor 488 (Invitrogen, cat. no.: A-11006, 1:200). Sections were counterstained with DAPI (4′,6-diamidino-2-phenylindole) (Sigma), and mounted using Prolong Diamond (Thermo Fisher). For quantitation of iNKT cell positioning in thymus sections, whole thymus tissue was incubated with phycoerythrin-conjugated PBS57-loaded CD1d tetramers overnight, and then subsequently fixed in 2% paraformaldehyde (PFA) (Sigma) overnight before freezing. Sections were incubated with rabbit anti-PE (Invitrogen, cat. no.: 12-4739-81, 1:100), and then further amplified using donkey anti-rabbit Alexa Fluor 555 (Invitrogen, cat. no.: A-31572, 1:1000). For quantitation of iNKT cell positioning, iNKT cells were detected in thymus sections by CD1d tetramer labelling and enumerated in 100 μm × 100 μm areas in the cortex and medulla. At least three medullary and three cortical regions were analysed per section. Four sections were analysed per mouse, obtained from differing depths throughout the thymus, and three mice of each genotype were analysed. Detection of CCL21[tdTOM] in sections was achieved by fixing whole thymus in 2% PFA/15% sucrose (Sigma) for 4 h prior to freezing. Cryosections were stained without further amplification of

the CCL21[tdTOM] signal. Images were obtained using a Zeiss Zen 880 microscope, and analysis was performed using the Zeiss Zen Black software.

**Foetal thymus organ culture.** Freshly isolated embryonic day 15 thymus lobes were isolated from B6 embryos generated by timed matings, with the day of detection of vaginal plug designated as day 0. Lobes were explanted onto the surface of 0.8 μm Nucleopore filters (Thermo Fisher) and cultured in the presence of 1.35 mM dGuo (Sigma)[54]. After 7 days, lobes were cultured for a further 4 days in the presence or absence of 2 μg/ml agonist anti-LTβR[24], and then digested with 0.25% trypsin and 0.02% EDTA for either flow cytometric staining or qPCR analysis.

**Quantitative polymerase chain reaction.** Briefly, high-purity complementary DNA (cDNA) was obtained from mRNA labelled with oligo(dT) microbeads using the μMacs One-step cDNA Kit, according to the manufacturer's instructions (Miltenyi Biotec). Real-time PCR was performed with SYBR Green with primers specific for *Actb* (β-actin), *Pou2f3*, *Dclk1*, *Trpm5*, *Ccl21a*, *Il25*, *Il15* and *Il15ra* on the Corbett Rotor Gene-3000 PCR machine (Qiagen). PCRs were ran in replicates using SensiMix SYBR No ROX Kit (Bioline) with 200 nM of primers for the genes of interest (Sigma-Merck) and *Actb* primers (Qiagen, QT00095242). After an initial polymerase activation step (95 °C for 10 min), cycling was performed for 40 cycles at 95 °C for 15 s, 60–62 °C for 20 s, and 72 °C for 15 s. Amplification specificity was verified by melt-curve analysis. Reaction amplification efficiencies and the Ct values were obtained from the Rotor Gene Real-Time Analysis Software 6.1 using standard curves generated from mouse Universal cDNA Reference-oligo dT primed (BioChain Institute). Fold levels shown in histograms represent the mean (±SEM) of replicate reactions and data shown are representative of at least three independently sorted sample sets. Primer sequences used:

*Pou2f3*: forward 5′-CTGGAACAGTAACGTCATCCTG-3′ and reverse 5′-AGTTCATTGCTGCTTTGGAGTT-3′;
*Dclk1*: forward 5′-GTTCCGTGGAAGTGGGGATG-3′ and reverse 5′-GCTATTACAGAAACTCCTGCTGC-3′;
*Trpm5*: forward 5′-CCAGCATAAGCGACAACATCT-3′ and reverse 5′-GAGCATACAGTAGTTGGCCTG-3′;
*Ccl21a*: forward 5′′-ATCCCGGCAATCCTGTTCTC-3′ and reverse 5′-GGGGCTTTGTTTCCCTGGG-3′;
*Il25*: forward 5′-TATGAGTTGGACAGGGACTTGA-3′ and reverse 5′-TGGTAAAGTGGGACGGAGTTG-3′;
*Il15*: forward 5′-TCTCCCTAAAACAGAGGCCAA-3′ and reverse 5′-TGCAACTGGGATGAAAGTCAC-3′;
*Il15ra*: forward 5′-CCCACAGTTCCAAAATGACGA-3′ and reverse 5′-GCTGCCTTGATTTGATGTACCAG-3′.
*Actb* (β-actin): QuantiTect Mm *Actb* 1SG Primer Assay (Qiagen, QT00095242).

**αGal-Cer stimulation and flow cytometric cytokine detection.** Mice were injected intravenously with either 2 μg αGal-Cer (Abcam) or PBS, and harvested 2 h later for detection of IL-4 and IFNγ by flow cytometry and serum IL-4. For serum detection of IFNγ, mice were harvested 16 h following αGal-Cer injection. Anti-IL-4 (11B11, eBioscience, cat. no.: 17-7041-82, 1:100) and anti-IFNγ (XMG1.2, BioLegend, cat. no.: 505813, 1:100) antibodies were used in conjunction with a Cytofix/Cytoperm Kit (Becton Dickinson).

**Serum cytokine detection.** Mouse IL-4 and IFN-γ ELISA Deluxe Kits (BioLegend) were used for quantification of cytokines in serum following in vivo αGal-Cer stimulation. Samples were diluted for optimal detection.

**In vitro cytokine stimulation.** Enrichment of thymic iNKT cells was achieved using anti-CD8 microbeads and LS columns (Miltenyi Biotec). Cells were cultured in the presence of rIL-25 (BioLegend) or IL-15/IL-15Rα complexes, made by mixing rIL-15 (Peprotech) and rIL-15Rα (R&D Systems) and incubating at 37 °C for 30 min. For in vitro assays, the final concentrations were: rIL-25, 10 ng/ml; rIL-15, 50 ng/ml; rIL-15Rα, 120 ng/ml.

**In vivo cytokine stimulation.** Mice were given a single intraperitoneal injection of either PBS, rIL-25 (2.5 μg, BioLegend) or IL-15/IL-15Rα complexes (rIL-15: 2.5 μg, Peprotech), rIL-15Rα: 15 μg, R&D Systems), and tissues were harvested and analysed 4 days later.

**Statistical analysis.** The GraphPad Prism software was used for all statistical analysis. Unpaired two-tailed Student's t tests were used for all statistical analysis. Only P values < 0.05 were noted as significant. Nonsignificant differences were not specified. Error bars in all figures represent SEM.

**Reporting summary.** Further information on research design is available in the Nature Research Reporting Summary linked to this article.

## Data availability

The authors confirm that all data that support the findings of this study are available in the figures and supplementary figures contained in the paper.

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

## Acknowledgements

We thank Dr. Andrea Bacon and the Biomedical Services Unit at The University of Birmingham for expert animal husbandry. We also thank Drs. Alexei Tumanov, Klaus Pfeffer and Andrew McKenzie for mice, Carl Ware for agonistic anti-LTβR, National Institutes of Health Tetramer Facility for CD1d/PBS57 tetramers and Dr. Kristin Hogquist for methods to detect iNKT in tissue sections. We express our gratitude to Dr. Kendle Maslowski for sharing the methods for gut histology, and all lab members for helpful discussion and critical review of the manuscript. This work was supported by an MRC programme grant to G.A.

## Author contributions

G.A., W.E.J. and B.L. conceived the study and designed experiments. B.L., A.J.W., S.M.P., E.J.C. and K.D.J. performed experiments and analysed data. N.D.J., I.O. and Y.T. provided important reagents and advice, and interpretation of data. B.L. and G.A. wrote the manuscript with input from all authors.

## Competing interests

The authors have no competing interests.
