## [Peer Review File · Nature Communications]

Editorial Note: This manuscript has been previously reviewed at another journal that is not operating a transparent peer review scheme. This document only contains reviewer comments and rebuttal letters for versions considered at Nature Communications .

REVIEWERS' COMMENTS:

Reviewer #2 (Remarks to the Author):

Additional experimentation has been performed and all comments have been addressed by the authors

Reviewer #3 (Remarks to the Author):

The authors have improved the paper significantly. Apologies on behalf of the reviewers, including me, who asserted that the previous work (Miller et al., 2018) established that tuft cells influenced NKT2 cells in the thymus through IL-25. The authors are correct this was not established. However, Watarai et al did show that NKT2 cells were selectively diminished in *Il17rb*^{-/-} mice (PLoS Biology volume 10, 2012), so it might still be fair to consider this advance to be incremental. (The caveat is that IL-17RB can bind another IL-17 family cytokine, but regardless, this Watarai paper should be referenced).

Regarding the mouse controls, there aren't much data to suggest haploinsufficiency of CD1d, although there are some subtle changes in the NKT cell V β repertoire. Also, the point of using the *Ja18*^{-/-} mice would be to rule out effects due to CD1d reactive T cells with more diverse TCRs (type II NKT cells), which could affect thymic DCs, and functions of CD1d not related to interactions with T cells. This sophisticated research group could have been more rigorous in carrying out this set of experiments.

Reviewer #4 (Remarks to the Author):

Thank you for the opportunity to act as a mediating referee for the manuscript NCOMMS-20-02118A "Functional Diversity of Medullary Epithelial Cells Controls iNKT-cell Activity and Pool Size in The Thymus and Periphery". In brief, reviewer 1 raised two major concerns, both regarding the novelty of this work compared to previous publications (Miller et al., 2018 Nature, Bornstein et al., 2018 Nature and Lkhagvasuren et al., 2013 J Immunol). I am satisfied that the authors have addressed both concerns adequately and the work presented in this manuscript is novel and suitable for publication in Nature Communications. The addition of new data from *IL-25*^{-/-} mice strengthens the manuscript and provides further support that the development of NKT cells is regulated by IL-25 produced by thymic tuft cells.

Reviewer 2.

We thank the reviewer for stating all comments have been addressed.

Reviewer 3.

As requested, we now include the reference by Watarai PLoS Biol 2012. Regarding use of additional NKT-cell deficient mouse strains, we now include a reference for the suggested *Ja18*^{-/-} mouse strain, and also include discussion on how use of these mice may also be of interest in relation to studying the influence of NKT-cells and thymus medulla.

Reviewer 4.

We thank the reviewer for acting as a mediating referee, and we are very grateful to see that they think our study is novel, of interest, and worthy of publication in Nature Communications.